# Mixture of Experts for Audio-Visual Learning

**Ying Cheng**[1,2,3]    **Yang Li**[1,2]    **Junjie He**[1,2]    **Rui Feng**[1,2,3*]

[1]School of Computer Science, Fudan University
[2]Shanghai Key Laboratory of Intelligent Information Processing
[3]Shanghai Collaborative Innovation Center of Intelligent Visual Computing
{chengy18, fengrui}@fudan.edu.cn,
{liy23, hejj23}@m.fudan.edu.cn

## Abstract

With the rapid development of multimedia technology, audio-visual learning has emerged as a promising research topic within the field of multimodal analysis. In this paper, we explore parameter-efficient transfer learning for audio-visual learning and propose the Audio-Visual Mixture of Experts (*AVMoE*) to inject adapters into pre-trained models flexibly. Specifically, we introduce unimodal and cross-modal adapters as multiple experts to specialize in intra-modal and inter-modal information, respectively, and employ a lightweight router to dynamically allocate the weights of each expert according to the specific demands of each task. Extensive experiments demonstrate that our proposed approach *AVMoE* achieves superior performance across multiple audio-visual tasks, including AVE, AVVP, AVS, and AVQA. Furthermore, visual-only experimental results also indicate that our approach can tackle challenging scenes where modality information is missing. The source code is available at https://github.com/yingchengy/AVMOE.

## 1   Introduction

Audio-visual learning is a challenging problem that analyzes auditory and visual information simultaneously, which aims at naturally integrating information from multiple modalities to perceive and understand the real-world comprehensively. By leveraging both auditory and visual cues, the models can learn multimodal correlations and solve various tasks such as Audio-Visual Event localization (AVE) [54], Audio-Visual Segmentation (AVS) [65], and Audio-Visual Question Answering (AVQA) [61, 27, 64].

In recent years, audio-visual learning has received steadily increasing attention in the field of multimodal analysis. Some previous works [3, 9, 22, 43, 1] utilize the correspondence between auditory and visual streams to learn multimodal representations in a self-supervised manner. By reducing the reliance on labeled data, self-supervised approaches can scale to larger datasets [14] and improve generalization performance. However, due to the rapid growth of the model size, fine-tuning full parameters of large pre-trained models entails significant computational costs for each downstream task. Recently, some researchers [30, 11] have attempted to inject adapters into frozen audio-visual pre-trained models, so as to reduce trainable parameters when applying to downstream tasks. Although these approaches have achieved satisfactory performance, they only focus on cross-modal attention in adapters to introduce information from other modalities, lacking the prominence of crucial information within the current modality. Besides, not every visual frame and audio segment can promote each other, and only introducing cross-modal adapters may bring irrelevant information during the learning process. As shown in Figure 1, the sound of basketball bounce is completely drowned out by the commentator's voice and the cheers from the audience. In these cases such

---

*Corresponding author

38th Conference on Neural Information Processing Systems (NeurIPS 2024).

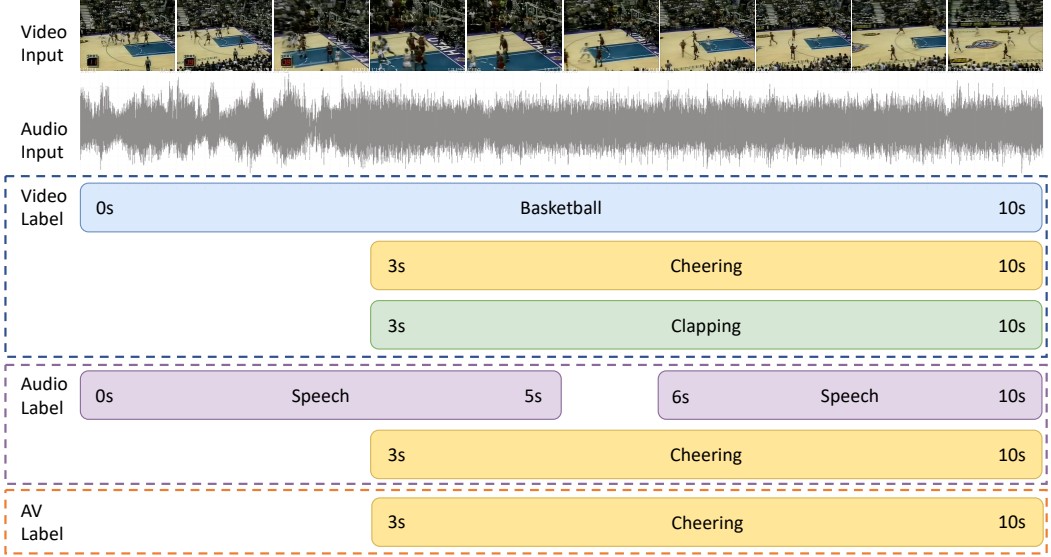

Figure 1: The case that video and audio labels are different. The video label includes *basketball*, *cheering*, and *clapping*, while the audio label includes *speech* and *cheering*.

as off-screen sounds, injecting cross-modal adapters only brings disturbing information, and it is necessary to inject adapters according to the scenario flexibly.

The Mixture of Experts (MoE) is a structural strategy for machine learning models, designed to enhance overall prediction performance by combining the decisions of multiple expert models. Hence, it is appealing to integrate the MoE module into neural networks. Within the MoE strategy, each expert model focuses on different subsets or aspects of the input data, providing predictions for a given input. Additionally, there is a gating mechanism (usually a trainable layer) responsible for determining which expert's predictions should be given more weight based on the input data. This structure allows the model to dynamically adjust its reliance on different experts, thereby optimizing its ability to process various types of data.

In this work, we propose an approach of **M**ixture **o**f **E**xperts for **A**udio **V**isual Learning (**AVMoE**), designed as an efficient and flexible approach to adaptively combine adapter capabilities according to different scenarios. Firstly, we introduce two types of adapters as experts into frozen pre-trained models, i.e., unimodal adapter and cross-modal adapter, to focus on intra-modal and inter-modal information, respectively. Secondly, we introduce a modality-agnostic router to evaluate how relevant each expert is to the characteristics of inputs and assign the weights to each expert. By employing the *AVMoE* scheme, the model can adapt its strategy according to the specific demands of each task, whether it be heavily reliant on visual data, audio, or requiring a balanced integration of both. We conduct extensive experiments on three audio-visual tasks, i.e., AVE, AVS, and AVQA. Experimental results demonstrate that our approach not only improves the model's performance across a wide range of audio-visual tasks but also makes it robust to variations in the input data. For instance, in scenarios where audio information is obscured or noisy, the model can shift its reliance toward visual cues and vice versa.

**The contributions of this paper can be summarized as follows:**

- We propose a generic, effective, and flexible approach, *AVMoE*, for applying pre-trained models to audio-visual learning, which can dynamically adjust its strategy according to the specific demands of each task.

- We introduce two types of adapters to focus on both within-modality details and the interactions between modalities. By integrating adapters and MoE, the model can be robust to any variations in the input data, e.g., missing modality or multimodal content mismatch.

- Extensive experiments demonstrate that our method consistently outperforms in diverse audio-visual tasks, showcasing its ability to dynamically prioritize visual or audio cues, enhancing its reliability and effectiveness in real-world applications.

# 2 Related Work

## 2.1 Audio-Visual Learning

Audio-visual learning aims to integrate auditory and visual information to enhance the understanding and perception of multimedia scenarios. Some early researchers [3, 22, 43] propose an innovative self-supervised approach that utilizes the correspondence between visual and audio components in video as a supervisory signal to pre-train a multimodal model, and the model can be applied to multiple downstream tasks such as speech enhancement [8, 17, 18], sound localization [60, 32], and action recognition [23, 49]. However, these audio-visual pre-trained models are limited by the ability of temporal localization and complex reasoning.

Recently, some researchers have made efforts for complicated audio-visual downstream tasks. Audio-Visual Event Localization (AVE [54]) aims at classifying and localizing events in the given video, and most prior works [57, 59, 62, 37] introduce interactions to utilize multimodal cues from modality-specific pre-trained models. Audio-Visual Video Parsing (AVVP [53]) expands the task of localizing one event to multiple events and removes the restriction that audio and visual events are definitely consistent. Previous works [63, 56] usually propose multi-scale hybrid networks and aggregate features in a weakly-supervised manner. Audio-Visual Segmentation (AVS [65]) aims to localize and segment the masks of sounding objects at the pixel level. To tackle this task, most prior methods learn correspondences between audio and visual patches and perform mask decoding by variational auto-encoder [39], contrastive conditional latent diffusion [38], vision transformer [31], bidirectional generation [15]. Audio-Visual Question Answering (AVQA [61]) requires the model to correctly answer the questions by understanding both audio and visual modalities comprehensively. Previous works tackle this task by spatiotemporal grounding model [26], missing modality recalling [44], and multi-scale feature fusion [7].

Although these researches have achieved satisfactory performance, designing models for distinct tasks separately is often expensive and time-consuming. In order to improve efficiency and generalization, some recent works have explored parameter-efficient transfer learning for audio-visual downstream tasks. LAVish [30] proposes to introduce cross-modal adapters into frozen pre-trained transformers for audio-visual data. DG-SCT [11] proposes dual-guided spatial-channel-temporal attention mechanisms as cross-modal prompts injected into pre-trained models. Unlike these prior methods focus on cross-modal adapters, we introduce uni-modal adapters and the MoE scheme to combine multimodal information dynamically.

## 2.2 Mixture of Experts

Mixture of Experts (MoE) [19] is a framework that has been extensively studied and applied in various domains of machine learning. Some works explore the role of MoE in the research fields of Natural Language Processing (NLP) [51, 24, 13, 10, 68] and Computer Vision (CV) [47, 35]. These works typically regard the Feed-Forward Network (FFN) layers as experts and choose the most relevant experts for the given task. Some efforts have been made to improve the routing strategy by differentiable selection [16], linear assignment [25], simple hashing algorithm [48], dense-to-sparse strategy [42], and expert choice [67].

Recently, some works have extended the original MoE scheme to enhance its applicability in various advanced learning paradigms. In the field of multi-task learning, Ma et al. [36] and Chen et al. [6] have successfully integrated MoE schemes to deal with the complexities of learning multiple tasks simultaneously. These approaches leverage expert-specific knowledge to effectively address the unique requirements of each task, thus improving the performance and efficiency of models.

Similarly, the MoE scheme has also been adapted for multimodal learning. Mustafa et al. [41] explore the usage of MoE in combining visual and textual information, demonstrating significant improvements in tasks that require understanding from both image and text inputs. Shen et al. [52] further employ MoE to scale both the text-based and vision-based feed-forward networks, showing the robustness and flexibility in handling diverse data sources. Cao et al. [4] and Akbari et al. [2] also contribute to this field by developing MoE-based models that alternate between different modalities, effectively capturing the complex interactions between them. However, these approaches only investigate the effectiveness of MoE in vision-language models, to the best of our knowledge, the MoE scheme has not been explored for audio-visual learning.

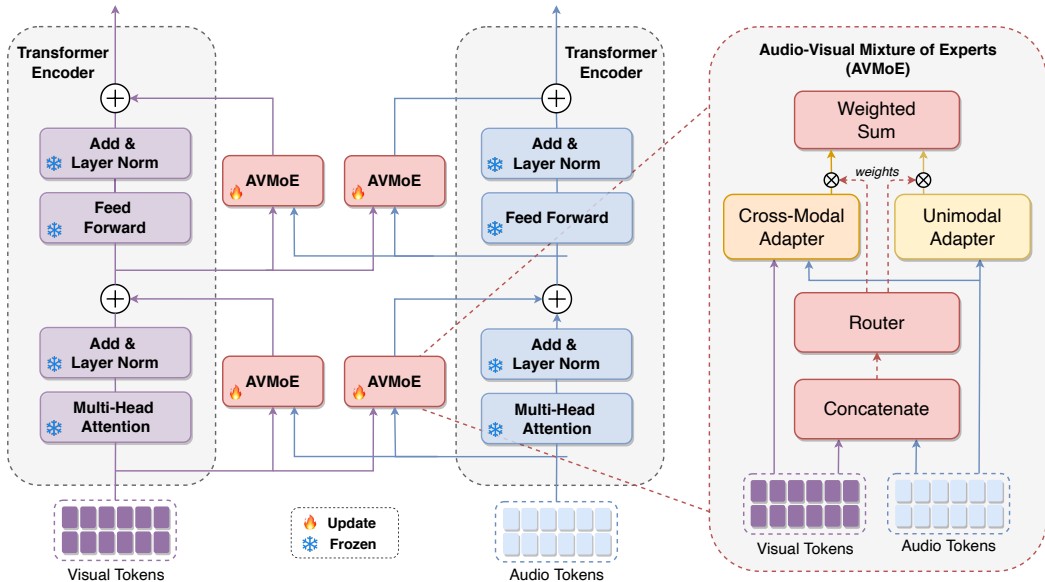

Figure 2: Method Overview. We propose to inject trainable adapters with the MoE scheme into the frozen pre-trained backbones for dynamically adjusting the strategy. The AVMoE module mainly consists of a router layer and two types of adapters, the router ingests concatenated multimodal tokens and allocates the weights for adapters, and the output of AVMoE module is the weighted sum of all the adapters' predictions.

## 3 Method

In this section, we present our approach *AVMoE* as shown in Figure 2, which adaptively integrates multiple types of adapters into the frozen pre-trained model using the audio-visual mixture of experts (MoE) framework. The structure of our framework consists of the frozen pre-trained model (Sec. 3.1) and trainable MoE modules (Sec. 3.2), where the MoE module contains two types of adapters and a router. In the following, we will describe each component in more detail.

### 3.1 Frozen Pre-trained Model

In this work, we aim to adapt pre-trained models to audio-visual tasks with minimal changes to the original parameters. Hence, we add a few trainable parameters while keeping the pre-trained model frozen. Following previous work [11], we use the standard Swin-Transformer (Swin-T) [33] and audio transformer (HTS-AT) [5] as our pre-trained backbones. Each layer of these models consists of Multi-Head Attention (MHA), Feed Forward Network (FFN), and Layer Normalization with Residual Connections (Add & Layer Norm).

### 3.2 Audio-Visual Mixture of Experts

In this subsection, we describe how our proposed *AVMoE* can flexibly adjust frozen pre-trained models to various audio-visual downstream tasks. As illustrated in the right of Figure 2, the AVMoE module contains three primary components: two types of adapters as experts, each specializing in different aspects of the multi-modal data, and a router for ingesting concatenated tokens and allocating the corresponding weights of experts.

#### 3.2.1 Router

The router layer is responsible for activating the appropriate experts based on the input features. To be specific, after obtaining visual embedding $v_t$ and audio embedding $a_t$, we concatenate them as the input $i_t$ of the AVMoE router. Following previous works [13], we employ Multi-Layer Perceptrons (MLPs) as the architecture of the router $R$, which produces weights by using the softmax function. The weights for Cross-Modal Adapter (CMA) and Unimodal Adapter (UA) are computed as follows:

$$w_{\text{CMA}} = \frac{\exp\left(r_{\text{CMA}}(i_t)\right)}{\exp\left(r_{\text{CMA}}(i_t)\right) + \exp\left(r_{\text{UA}}(i_t)\right)}, w_{\text{UA}} = \frac{\exp\left(r_{\text{UA}}(i_t)\right)}{\exp\left(r_{\text{CMA}}(i_t)\right) + \exp\left(r_{\text{UA}}(i_t)\right)} \tag{1}$$

where $r_{\text{CMA}}$ and $r_{\text{UA}}$ are the outputs of the router for CMA and UA. The produced weights determine the importance of each adapter's prediction to the final output.

To promote the balanced usage of experts, inspired by previous works [13, 20], we introduce Gaussian noise to the router's output at the training stage, thereby promoting a more balanced use of experts. This method prevents the model from over-relying on certain experts, forcing the model to explore a wider range of experts:

$$g' = g + \epsilon, \quad \epsilon \sim \mathcal{N}(0, \sigma^2) \tag{2}$$

where $\epsilon$ is Gaussian noise with a standard deviation $\sigma$.

### 3.2.2 Audio-Visual Adapters

To enhance the efficiency and performance of audio-visual models, we follow previous works [30, 11], adding a few trainable parameters (i.e., adapters) into each layer of frozen pre-trained backbones. However, different from them, we design two types of adapters, i.e., CMA and UA. As shown in Figure3, the main differences between them are multimodal feature fusion and self-attention. These adapters are injected into the frozen pre-trained model via the MoE strategy.

**Cross-Modal Adapter.** Firstly, to introduce interactions and integrate information from multiple modalities, we follow *LAVisH* adapter [30] as our cross-modal adapter. As shown in the left of Figure3, the cross-modal adapter consists of audio-visual latent token compression, audio-visual feature fusion, and bottleneck module.

We propose to utilize latent tokens, which are small, randomly initialized vectors that serve as placeholders for condensed information from the audio and visual modalities. For layer $l$, we have $m$ latent tokens for audio, denoted as $L_a^l$, and $m$ latent tokens for visual, denoted as $L_v^l$. These tokens are significantly fewer in number compared to the total number of tokens from the audio or visual inputs, which helps in reducing the computational complexity.

In this step, the adapter uses cross-modal attention to compress the information from all the audio or visual tokens into the corresponding latent tokens. This is achieved by applying the cross-modal attention operation between the input tokens and the latent tokens of the same modality. The compressed latent tokens capture the most relevant information from the original tokens, which is essential for efficient information transfer between modalities.

For audio and visual inputs, the compression is given by:

$$S_a^l = f_c(L_a^l, X_a^l, X_a^l), \quad S_v^l = f_c(L_v^l, X_v^l, X_v^l) \tag{3}$$

where $f_c$ is the cross-attention function, $X_a^l$ and $X_v^l$ are the audio and visual tokens at layer $l$, and $S$ represents the resulting latent summary tokens.

After compressing the modality-specific information into latent tokens, the adapter fuses the compressed information from one modality with the full set of tokens from the other modality. This is done through another round of cross-modal attention, which allows the model to integrate audio and visual cues effectively. For cross-modal fusion:

$$X_{av}^l = f_c(X_a^l, S_v^l, S_v^l), \quad X_{va}^l = f_c(X_v^l, S_a^l, S_a^l) \tag{4}$$

where $X_{av}^l$ and $X_{va}^l$ represent the newly computed audio and visual representations that incorporate information from both modalities.

The final step in the adapter involves a lightweight module that refines the fused audio-visual representations into more discriminative features suitable for downstream tasks. This module consists of a bottleneck structure with a down-projection layer ($\theta^{down}$), a non-linear activation function ($\sigma$), and an up-projection layer ($\theta^{up}$). For the audio-visual and visual-audio pathways:

$$Z_{av}^l = \theta^{up}(\sigma(\theta^{down}(X_{av}^l))), \quad Z_{va}^l = \theta^{up}(\sigma(\theta^{down}(X_{va}^l))) \tag{5}$$

where $Z_{av}^l$ and $Z_{va}^l$ are output features that are passed to subsequent layers or used for task-specific predictions.

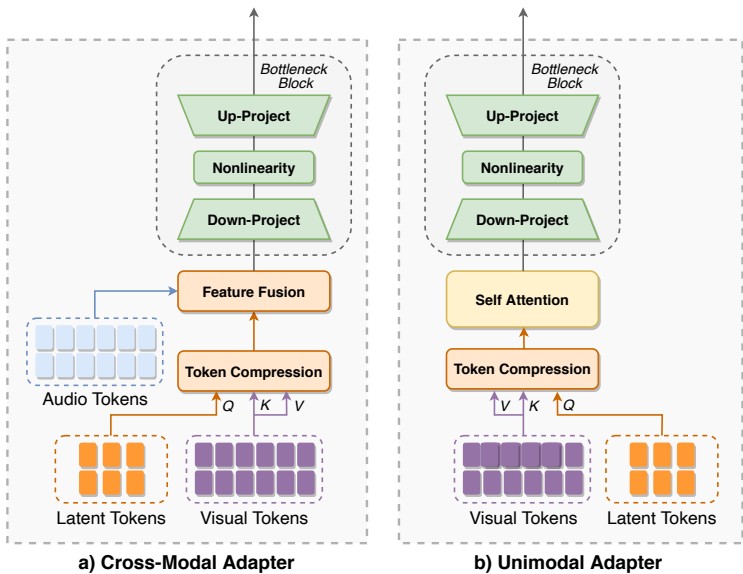

Figure 3: The structure of adapters. a) Cross-Modal Adapter (CMA), consisting of token compression, multimodal feature fusion and bottleneck block; b) Unimodal Adapter (UA), consisting of token compression, self-attention and bottleneck block.

**Unimodal Adapter.** While cross-modal adapters effectively introduce interactions between audio and visual tokens, they may not always be suitable for scenarios where auditory and visual content do not perfectly align, potentially introducing irrelevant information. Additionally, some multimodal data might primarily rely on one modality, or another modality may be completely missing, making inter-modal interactions less effective. Hence, it is also necessary to introduce intra-modal interactions and design unimodal adapters, which can address these issues by providing targeted improvements that compensate for the limitations of cross-modal adapters. As shown in the right of Figure3, after obtaining compressed tokens $S_a^l$ and $S_v^l$, the self-attention layer ingests them and outputs newly computed representations.

$$X_a^l = f_s(X_a^l, S_v^l, S_v^l), \quad X_v^l = f_s(X_v^l, S_a^l, S_a^l) \tag{6}$$

where $f_c$ is the cross-attention function, $X_a^l$ and $X_v^l$ represent the output audio and visual representations, respectively.

In this work, we propose to utilize MoE strategy to introduce intra- and inter-modal interactions synergistically. By integrating these two types of adapters, our model can dynamically adjust the attention paid to unimodal and cross-modal information according to the scenario.

## 4 Experimental Analysis

To demonstrate the capabilities of our proposed *AVMoE* in solving diverse audio-visual tasks, we conduct extensive experiments on four downstream tasks, i.e., Audio-Visual Event Localization (AVE), Audio-Visual Video Parsing (AVVP), Audio-Visual Segmentation (AVS), and Audio-Visual Question Answering (AVQA). More details about audio-visual tasks and datasets will be illustrated in the Appendix.

### 4.1 Audio-Visual Event Localization

For the AVE task, we adopt the overall segment-wise accuracy of predicted event categories as the evaluation metric, and the event label of each video segment is required to be predicted in a fully-supervised manner. As shown in Table 1, we compare our proposed *AVMoE* with previous methods on the test set of AVE dataset. We focus on *LAViSH* [30] and *DG-SCT* [11], as they also explore parameter-efficient fine-tuning methods like ours and have achieved impressive results on this benchmark.

Table 1: **Audio-Visual Event Localization.** Comparison with previous methods in a fully-supervised manner. The performance is evaluated on the test set of AVE dataset with classification accuracy. The visual and audio encoders are pre-trained on the ImageNet and Audioset, respectively. † means that no official code was provided to report some of the baseline-specific metrics.

| Method | Visual Encoder | Audio Encoder | Trainable Params (%) ↓ | Total Params (M) ↓ | Acc ↑ |
|---|---|---|---|---|---|
| AVT [29] | VGG-19 | VGGish | 6.8 | 231.5 | 76.8 |
| MPN [62] | VGG-19 | VGGish | N/A | N/A | 77.6 |
| PSP [66] | VGG-19 | VGGish | 0.8 | 217.4 | 77.8 |
| DPNet† [46] | VGG-19 | VGGish | N/A | N/A | 79.7 |
| AVEL [54] | ResNet-152 | VGGish | 2.7 | 136.0 | 74.0 |
| AVSDN [28] | ResNet-152 | VGGish | 5.7 | 140.3 | 75.4 |
| CMRAN [59] | ResNet-152 | VGGish | 10.7 | 148.2 | 78.3 |
| MM-Pyramid [63] | ResNet-152 | VGGish | 25.0 | 176.3 | 77.8 |
| CMBS [58] | ResNet-152 | VGGish | 6.6 | 216.7 | 79.7 |
| LAVisH [30] | ViT-B-16 (shared) | | 4.4 | 107.2 | 75.3 |
| LAVisH [30] | ViT-L-16 (shared) | | 4.3 | 340.1 | 78.1 |
| AVMoE (Ours) | ViT-B-16 (shared) | | 31.8 | 150.4 | 76.4 |
| AVMoE (Ours) | ViT-L-16 (shared) | | 32.6 | 483.1 | 79.2 |
| LAVisH [30] | Swin-V2-B (shared) | | 4.4 | 114.2 | 78.8 |
| LAVisH [30] | Swin-V2-L (shared) | | 2.7 | 238.8 | 81.1 |
| **AVMoE (Ours)** | Swin-V2-B (shared) | | 41.1 | 206.6 | 79.4 |
| **AVMoE (Ours)** | Swin-V2-L (shared) | | 39.4 | 374.4 | 81.5 |
| LAVisH [30] | Swin-V2-L | HTS-AT | 30.6 | 374.9 | 78.6 |
| DG-SCT [11] | Swin-V2-L | HTS-AT | 43.6 | 461.3 | 82.2 |
| **AVMoE (Ours)** | Swin-V2-L | HTS-AT | 34.9 | 404.0 | **82.6** |

For a fair comparison, we train our *AVMoE* with shared pre-trained visual backbones (ViT-B, ViT-L, Swin-V2-B, Swin-V2-L) and separate audio-visual backbones (HTS-AT and Swin-V2-L). Firstly, it can be seen that our method achieves consistently better performance than *LAVisH* across different shared backbones, which indicates that our method can be adapted to different backbones. We note that the structure of Swin-V2-L is better than ViT, since Swin-V2-L allows hierarchical feature representation and more efficient computation by constraining self-attention within local windows and progressively expanding them. Secondly, when utilizing distinct pre-trained backbones for each modality, our model outperforms *LAVisH* and *DG-SCT* by 4.0% and 0.4%, respectively. It should be noted that the number and proportion of trainable parameters is less than that of *DG-SCT* with complicated attention mechanisms, showcasing the efficiency and superiority of our model. Lastly, we also observe that the audio encoder pre-trained on Audioset can further improve the performance, which indicates that combining different modalities is beneficial for audio-visual learning.

### 4.2 Audio-Visual Video Parsing

Compared to the AVE task, the AVVP task is a more complex and challenging multimodal task involving analyzing and understanding audio and visual information within a video to identify and categorize events. This task aims to segment a video into temporal events and predict their modality-specific categories, which may include actions that are audible, visible, or both. A key challenge in AVVP is that audio and visual modalities may not be fully correlated. For instance, due to limitations in camera field-of-view or occlusions, some events may occur in only one modality. Therefore, effectively integrating audio and visual information while mitigating the interference of irrelevant modality information is crucial.

As shown in Table2, our *AVMoE* has significant advantages over *DG-SCT* and *MGN* on the AVVP task. For example, the audio-visual event parsing metric at segment-level exceeds DG-SCT by 2%, which indicates that the scheme of MoE is suitable for balancing weight assignment for multi-modal information.

Table 2: **Audio-Visual Video Parsing.** Comparison with previous methods on the test set of LLP dataset. We conduct comparative experiments on two kinds of annotation: segment-level and event-level. The **Type** computes the average audio, visual, and audio-visual event evaluation results, and the **Event** considers all audio and visual events for each sample to compute the F-score.

| Method | Segment-level | | | | | Event-level | | | | |
|---|---|---|---|---|---|---|---|---|---|---|
| | A | V | AV | **Type** | **Event** | A | V | AV | **Type** | **Event** |
| AVE[54] | 49.9 | 37.3 | 37.0 | 41.4 | 43.6 | 43.6 | 32.4 | 32.6 | 36.2 | 37.4 |
| AVSDN[28] | 47.8 | 52.0 | 37.1 | 45.7 | 50.8 | 34.1 | 46.3 | 26.5 | 35.6 | 37.7 |
| HAN[53] | 60.1 | 52.9 | 48.9 | 54.0 | 55.4 | 51.3 | 48.9 | 43.0 | 47.7 | 48.0 |
| MGN[40] | 60.7 | 55.5 | 50.6 | 55.6 | 57.2 | 51.0 | 52.4 | 44.4 | 49.3 | 49.2 |
| DG-SCT[11] | 59.0 | 59.4 | 52.8 | 57.1 | 57.0 | 49.2 | **56.1** | 46.1 | 50.5 | 49.1 |
| **AVMoE(Ours)** | **62.1** | **60.0** | **54.4** | **58.8** | **59.0** | **51.8** | 55.7 | **47.6** | **51.7** | **50.2** |

Table 3: **Audio-Visual Segmentation.** Comparison with previous methods under the S4 and MS3 settings of AVSBench dataset. $\mathcal{M_J}$ and $\mathcal{M_F}$ denote the mean $\mathcal{J}$ and $\mathcal{F}$ metric values over the whole dataset.

| Method | Visual Encoder | Audio Encoder | Trainable Params (%) ↓ | Total Params (M) ↓ | Setting | | | |
|---|---|---|---|---|---|---|---|---|
| | | | | | S4 | | MS3 | |
| | | | | | $\mathcal{M_J}$ | $\mathcal{M_F}$ | $\mathcal{M_J}$ | $\mathcal{M_F}$ |
| AVS [65] | PVT-v2 | VGGish | 58.7 | 174.5 | 78.7 | 87.9 | 54.0 | 64.5 |
| LAVisH [30] | Swin-V2-L (shared) | | 14.0 | 266.4 | 80.1 | 88.0 | 49.8 | 60.3 |
| LAVisH [30] | Swin-V2-L | HTS-AT | 47.6 | 389.7 | 78.0 | 87.0 | 49.1 | 59.9 |
| DG-SCT [11] | Swin-V2-L | HTS-AT | 61.5 | 594.8 | 80.9 | 89.2 | 53.5 | 64.2 |
| **AVMoE (ours)** | Swin-V2-L | HTS-AT | 54.4 | 501.2 | **81.1** | **89.7** | **54.5** | **68.7** |

## 4.3 Audio-Visual Segmentation

To explore whether the model can associate the visual regions with the sound components, we compare our approach with state-of-the-art models on the AVS task. The Jaccard index $\mathcal{J}$ [12] and F-score $\mathcal{F}$ are used as the evaluation metrics, where $\mathcal{J}$ and $\mathcal{F}$ denote the region similarity and contour accuracy, respectively. As illustrated in Table 3, our approach *AVMoE* significantly outperforms *DG-SCT* and *LAVisH* under both S4 and MS3 settings, which verifies that the scheme of MoE is also beneficial for audio-visual segmentation. Our model shows more significant improvements under the MS3 setting of multi-sound sources, indicating that the model with *AVMoE* strategy can deal with more complex scenarios.

Figure 4 shows the qualitative results of our *AVMoE* model and *DG-SCT* on the AVS task. As can be seen, under the S4 setting, our model locates and segments the sounding objects more accurately than *DG-SCT*. For the case of ambulance siren (column #1), *DG-SCT* mistakenly locates the ambulance and the car nearby at the same time, while our model can exclude the car that hasn't produced sound. For the case of dog barking (column #3), our *AVMoE* model contours the object shapes (e.g., tail) more perfectly. Under the MS3 setting, compared to *DG-SCT* which cannot locate all sound sources in some cases, we can almost locate and segment each sound source. It can be seen that even in the case of three sound sources (column #4), we can still segment each instrument well, which verifies the effectiveness of our *AVMoE* model.

## 4.4 Audio-Visual Question Answering

We also want to explore whether our model can generalize to complex audio-visual tasks. To this end, we conduct experiments on the AVQA task and compare our model with previous leading methods. The AVQA task contains three categories of questions (audio, visual, and audio-visual), which involve questions related to audio (AQ), visual (VQ), or both audio and visual information simultaneously (AVQ). The multiple forms of questions challenge the adaptability to various modalities and the spatio-temporal reasoning ability of each model.

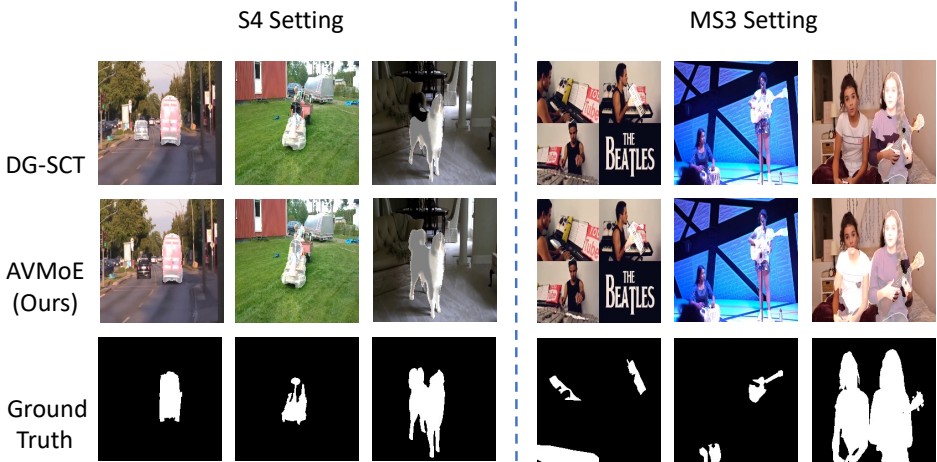

Figure 4: Qualitative examples of *AVMoE* and *DG-SCT* under the S4 setting and MS3 setting of the AVS task.

Table 4: **Audio-Visual Question Answering.** Comparison with previous methods on the test set of MUSIC-AVQA dataset. We report accuracy on three types of questions, i.e., Audio Question (AQ), Visual Question (VQ), and Audio-Visual Question (AVQ). LAVisH* denotes our implementation version of LAVisH.

| Method | Visual Encoder | Audio Encoder | Trainable Params (%)↓ | Total Params (M)↓ | AQ | VQ | AVQ | Avg |
|---|---|---|---|---|---|---|---|---|
| AVSD [50] | VGG-19 | VGG-like | N/A | N/A | 68.5 | 70.8 | 65.5 | 67.4 |
| Pano-AVQA [64] | Faster RCNN | VGG-like | N/A | N/A | 70.7 | 72.6 | 66.6 | 68.9 |
| ST-AVQA [27] | ResNet-18 | VGG-like | 11.2 | 94.4 | 74.1 | 74.0 | 69.5 | 71.5 |
| LAVisH [30] | Swin-V2-L(shared) | | 8.4 | 249.8 | 75.7 | 80.4 | 70.4 | 74.0 |
| LAVisH* [30] | Swin-V2-L | HTS-AT | 28.7 | 367.4 | 75.4 | 79.6 | 70.1 | 73.6 |
| DG-SCT [11] | Swin-V2-L | HTS-AT | 50.0 | 520.2 | 77.4 | 81.9 | 70.7 | 74.8 |
| **AVMoE (ours)** | Swin-V2-L | HTS-AT | 42.7 | 456.6 | **77.6** | **82.7** | **71.9** | **75.7** |

Table 4 reports the experimental results on the Music-AVQA dataset[27] across three distinct scenarios. Notably, our method exhibits superior generalization abilities and consistently achieves State-Of-The-Art (SOTA) performance in all scenarios. Especially in the challenging scenarios of AVQ, our *AVMoE* outperforms *DG-SCT* by 1.2% (71.9% v.s. 70.7%)with fewer trainable and total parameters, which verifies the effectiveness and efficiency of our approach. The dynamic adaptability of MoE strategy not only establishes the robustness of audio-visual models but also illustrates the potential of leveraging the flexible strategy to enhance performance in complex audio-visual tasks. The comprehensive experimental results across four audio-visual downstream tasks illustrate that our approach achieves significant improvements while efficiently transferring from frozen backbones to various downstream tasks.

## 5 Ablation Studies

### 5.1 Number of Experts

As shown in Table 5, we first investigate the impact of the number of experts (i.e., cross-modal adapters and uni-modal adapters) on the tasks of AVE, AVS, and AVQA. It can be seen that increasing the number of experts consistently enhances the model's performance, thereby validating the effectiveness of the MoE strategy. Besides, we observe that the performance of our model with only one CMA and one UA still exceeds that of *LAVisH*. Especially under the MS3 setting of Audio-Visual Segmentation (AVS) task, it has improved by 7.2% compared with *LAVisH* (67.5% v.s. 60.3%). We conjecture that introducing more experts may further improve the abilities of the model. However, due to the limitation of GPU memory, we did not discuss the results in this work.

Table 5: **AVMoE design.** CMA and UA denote cross-modal adapter and uni-modal adapter, respectively.

| Num. Experts | | AVS (S4) | | AVS(MS3) | | AVQA (Acc.) | | | | AVE (Acc.) |
|---|---|---|---|---|---|---|---|---|---|---|
| CMA | UA | $\mathcal{M}_{\mathcal{J}}$ | $\mathcal{M}_{\mathcal{F}}$ | $\mathcal{M}_{\mathcal{J}}$ | $\mathcal{M}_{\mathcal{F}}$ | AQ | VQ | AVQ | Avg | |
| 1 | 1 | 80.2 | 88.6 | 51.8 | 67.5 | 75.7 | 81.4 | 70.9 | 74.7 | 81.3 |
| 1 | 2 | 80.6 | 89.1 | 53.1 | 68.1 | 76.3 | 82.0 | 71.4 | 75.1 | 81.8 |
| 2 | 1 | 80.8 | 89.2 | 53.6 | 68.4 | 76.2 | 82.1 | 71.6 | 75.1 | 82.3 |
| 2 | 2 | **81.1** | **89.7** | **54.5** | **68.7** | **77.6** | **82.7** | **71.9** | **75.7** | **82.6** |

Table 6: **Modality Ablation.** "A" and "V" denote audio and visual, respectively. All models are well-trained on audio-visual modalities and tested on different modalities.

| Methods | Modalities | AVE(Acc.) | AVS($\mathcal{M}_{\mathcal{F}}$) | AVQA(Acc.) |
|---|---|---|---|---|
| DG-SCT | V | 72.1 | 79.1 | 67.3 |
| | A+V | 82.2 | 89.2 | 74.8 |
| AVMoE (Ours) | V | 78.4 | 85.4 | 73.2 |
| | A+V | 82.6 | 89.7 | 75.7 |

## 5.2 Modality Ablation

In order to explore how our well-trained *AVMoE* model handles different modality inputs, we conduct experiments of our model and *DG-SCT* on visual-only data and audio-visual data. As shown in Table 6, we compare our *AVMoE* model with *DG-SCT* on AVE, AVS, and AVQA tasks. It can be seen that our approach achieves significant performance on the test scenarios of visual-only data and audio-visual data. We observe that, our *AVMoE* model, when compared to *DG-SCT*, there is no significant decrease in accuracy on visual-only data. These results indicate that our model with the MoE scheme can still achieve good performance even in the absence of audio information. Although *DG-SCT* designs a more complex architecture of dual-guided spatial-channel-temporal attention mechanism in adapters, only introducing cross-modal adapters may bring irrelevant information in these challenging cases.

Moreover, to explore whether the model can dynamically activate experts, we visualize the activation weight of each expert varies across parallel expert layers. The activation ratio of experts is determined by calculating the ratio of each expert's selection frequency in each MoE layer to the total number of tokens. The visualization and more details will be discussed in Appendix.

## 6 Conclusion

In this paper, we develop a flexible framework that introduces Mixture of Experts (MoE) for audio-visual learning. Specifically, we inject unimodal adapters and cross-modal adapters as experts into the frozen pre-trained model and our *AVMoE* can dynamically combine the abilities of adapters according to different scenarios through the MoE strategy. Experimental results illustrate that our *AVMoE* has achieved greatly improved performance on complex tasks (such as AVQA and the MS3 setting of AVS). The ablation studies and qualitative results demonstrate that our model benefits significantly from the *AVMoE* strategy. However, due to computational resource limitations and efficiency considerations, we only investigate a small number of experts in the MoE scheme. In subsequent work, we will explore how to efficiently introduce more experts into our model.

## 7 Acknowledgements

This work was supported in part by the National Natural Science Foundation of China (No.62172101), in part by the Science and Technology Commission of Shanghai Municipality (No.23511100602), and supported by the Postdoctoral Fellowship Program of CPSF (No. GZC20230483).

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

# A  Audio-Visual Tasks and Datasets

To demonstrate the capabilities of our proposed *AVMoE* in solving diverse audio-visual tasks, we conduct extensive experiments on four downstream tasks, i.e., Audio-Visual Event Localization (AVE), Audio-Visual Video Parsing (AVVP), Audio-Visual Segmentation (AVS), and Audio-Visual Question Answering (AVQA). The transferring framework is shown in Figure 5.

## A.1  Audio-Visual Event Localization

AVE task is defined as an event that is both visible and audible in a video segment[54]. The AVE dataset focuses on combining audio and visual modalities to temporally mark the boundaries of all audio-visual events. It contains 4,143 videos covering 28 event categories, with each video lasting up to 10 seconds and including at least one 2s long audio-visual event. For this task, we first divide the video into 10 consecutive time segments. Then, we use the frozen transformers (e.g., ViT, Swin or HTS-AT) optimized by the AVMoE adapters to extract the audio and visual features that incorporate cross-modal interaction information from these segments. Last, we predict the audio-visual events by feeding the concatenated audio-visual features through two MLP layers, as shown in Figure 5. Referring to previous methods[54, 46, 58, 66], we calculate the proportion of time segments that are correctly predicted and use this as a standard to evaluate the performance of our method.

## A.2  Audio-Visual Video Parsing

AVVP task aims to parse videos into temporal event segments and label them as audible, visible, or both. For this task, we use the Look, Listen, and Parse (LLP) dataset[53], which contains 11,849 videos from different domains, covering 25 categories. LLP is a semi-supervised annotation dataset, and each video has video-level event annotations. Only 1,849 randomly selected videos have second-by-second annotations for audio and visual events. We made improvements based on MGN[40] by using *AVMoE* to enhance the Transformer layers in it, as shown in Figure5. Referring to previous methods[53, 40], F-scores are used to evaluate segment-level and event-level predictions of audio, visual, and audiovisual events which are predicted by our method.

## A.3  Audio-Visual Segmentation

AVS task aims to achieve pixel-level segmentation of visual objects based on audio. We conduct experiments on the AVSBench[65], a dataset containing both single-source segmentation and multiple-source segmentation. The single-source dataset contains 4,932 semi-supervised videos. During the training process, only the first sample frame of the video is fully labeled, but during evaluation, all video frames need to be predicted. The multi-source dataset contains 424 fully supervised videos with every frame labeled. For this task, we optimize the frozen transformer with trainable adapters to replace the pre-trained visual encoder and audio feature extractor of the U-Net used in AVS[65]. Then, the audio and visual features are fused and ingested into the decoder for prediction, as shown in Figure5. Referring to previous methods[65], we calculate the mean Intersection-over-Union (mIoU) of the predicted segmentation and the ground truth masks to evaluate our method.

## A.4  Audio-Visual Question Answering

AVQA task aims to answer questions about different visual objects, sound sources and their associations in the video. We conduct experiments on the MUSIC-AVQA dataset[27], which contains more than 45K question-answer pairs covering 33 different question templates. For this task, we utilize the trainable adapter to optimize the frozen Transformer and then replace the pre-trained visual and audio encoders of the baseline proposed in the AVQA dataset, as shown in Figure5. Referring to previous methods[27], the answer prediction accuracy is used to evaluate the performance of methods.

# B  Audio-Visual Embeddings

To enhance the fusion of audio-visual features, we follow the previous work [11], which utilizes pre-trained vision and audio transformers to encode visual and audio inputs, respectively. The encode process is as follows.

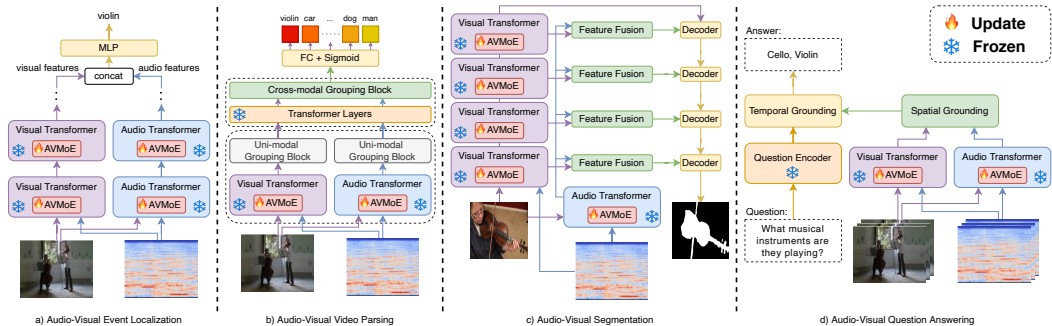

Figure 5: Applying AVMoE to audio-visual downstream tasks: audio-visual event localization, audio-visual video parsing, audio-visual segmentation, and audio-visual question answering. The purple and blue modules are frozen pre-trained visual and audio models, and the red modules are our proposed trainable adapter modules, and the remaining modules are set up with reference to the baseline model of these tasks, with some parameters trainable.

**Visual embedding.** For the given video sequence, we sample a set of RGB visual frames $\{V_t\}_{t=1}^T \in \mathbb{R}^{T \times H \times W \times 3}$, where $H$ and $W$ denote the height and width of frames, respectively. These video frames are sampled at the rate of 1 fps and then decomposed into $n$ non-overlapping patches. The patches are treated as individual tokens for sequence processing by the transformer model, and we can obtain the visual embeddings $v_t \in \mathbb{R}^{n \times d}$, where $d$ denotes the embedding dimension.

**Audio embedding.** For the given audio stream, we first employ Short-Time Fourier Transform (STFT) to convert waveforms into spectrograms $\{A_t\}_{t=1}^T \in \mathbb{R}^{T \times L \times F}$, where $L$ and $F$ denote the time and frequency bins, respectively. Similarly, the spectrograms are also divided into $k$ non-overlapping patches and then projected into audio embeddings $a_t \in \mathbb{R}^{k \times d}$.

## C  Implementation Details

For the above dataset, we extract visual frames at one frame per second and audio frames from video files at 8,000fps. We use the pre-trained Swin-V2-Large[34] as the visual transformer module with a spatial resolution of 192×192, and the pre-trained HTS-AT[5] as the audio transformer module. All of their parameters are frozen. We use the gradient accumulation method to accommodate the large parameters of the model, with weight updates after every 16 batches of training. For AVE and AVVP tasks, 32 latent tokens and a downsampling factor of 8 are used in the AVMoE adapter. For the AVS task on AVSBench and the AVQA task on MUSIC-AVQA, we use two latent tokens and set the downsampling factor and the number of group convolutions to 8 and 4, respectively.

For all of our experiments, we utilize the Adam[21] optimizer to train our models and set the scheduler to make the learning rate decay to 0.35 times its original value after every 3 epochs. We set the learning rate of the AVMoE adapter to 5e-4, while the learning rates of the final prediction layer are the same as previous work [11], 5e-6 for AVE, 3e-4 for AVVP, 3e-4 for the S4 setting of AVS, 1.5e-4 for the MS3 setting of AVS, and 1e-4 for AVQA. For audio pre-processing, we compute audio spectrograms via the PyTorch[45] kaldi fbank with 192 triangular mel-frequency bins and set frameshift over 5.2ms. For the task with audio-visual shared Transformer, we spliced the input channels of the audio spectrogram by tensor replication into 3 channels to match the input dimensions of the SwinV2 model.

For these audio-visual downstream tasks, all experiments are conducted on 8x NVIDIA 3090 (24G) GPUs, and the batch size on a single GPU varies depending on the parameters of the models. The one GPU batch size is set to be 1 for AVE and AVVP tasks, and 2 for AVS and AVQA tasks. Besides, we employ a gradient accumulation strategy to mitigate the computational resource limitations. To avoid overfitting, we set the early-stop hyper-parameterization to end the training process after five epochs with no improvement in model performance. Most of the experiments converge to the best results in about ten epochs.

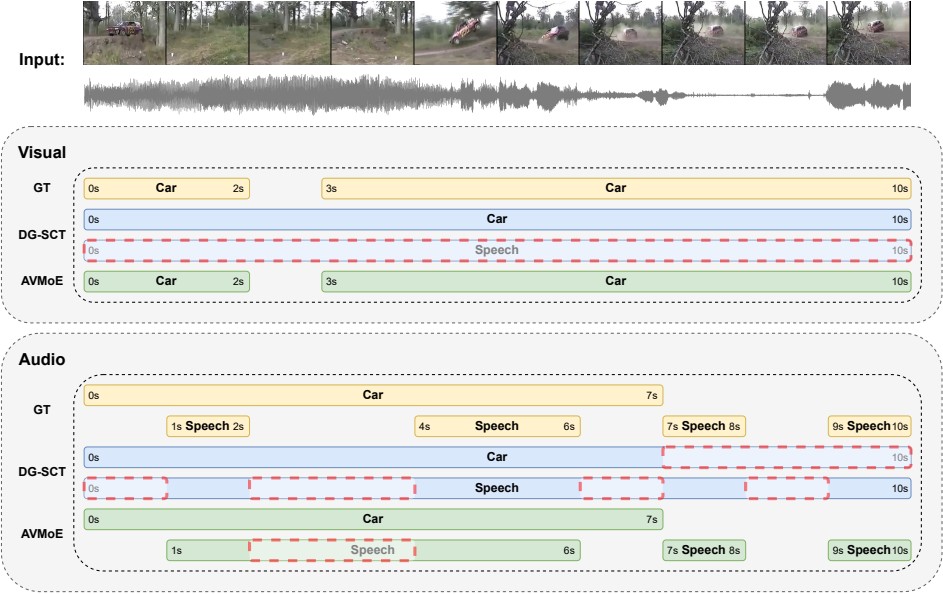

Figure 6: The qualitative experimental results on the AVVP task.

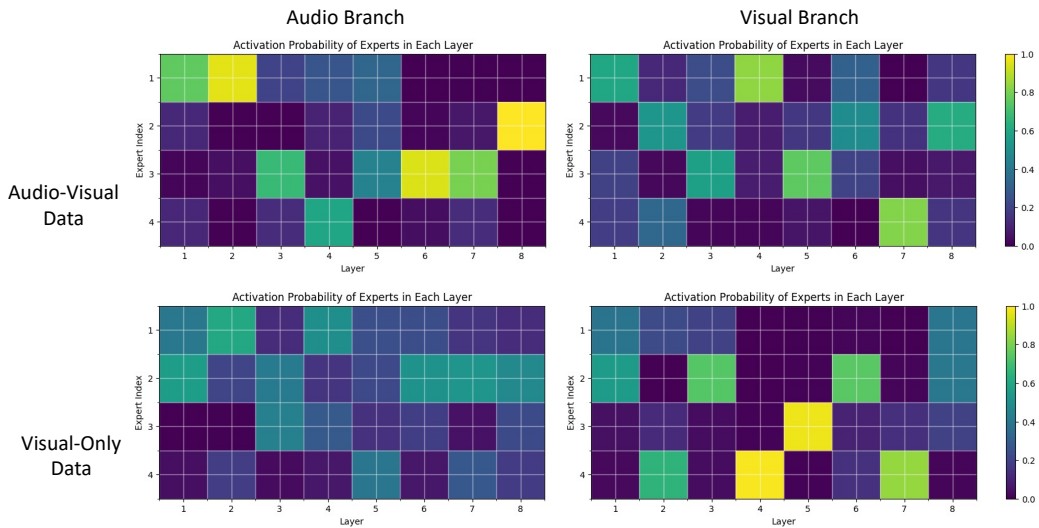

Figure 7: The activation probability of experts. Experts #1 and #2 denote cross-modal adapters, and experts #3 and #4 denote unimodal adapters.

## D  Qualitative Experiments of Audio-Visual Video Parsing

Figure 6 shows the qualitative experimental results on the AVVP task. In this example, the video label only includes car, and the audio label includes car and speech, which are not exactly the same. We observe that the visual predictions of *DG-SCT* are affected by the audio features, resulting in a visual prediction of speech, which does not exist in the video frame. On the other hand, our *AVMoE* retains unimodal information and has MoE modules to adjust the weight distribution for audio-visual features. This avoids the interference of inconsistent information and obtains the correct result. As for the audio predictions, our model can better localize the temporal boundaries of events compared to *DG-SCT*, which reveals that our model is capable of dealing with complex scenarios.

# E    The activation probability of experts

The activation visualization of experts is shown in Figure 7. It can be observed that: 1)In the visual branch, The distribution of expert activation is relatively even, while in the audio branch, the activation probability of the crossmodal adapter is higher. We hypothesize that this may be due to the visual information plays a vital role in the processing of audio-visual data. 2) When the model deals with visual-only data, the activation probability of the unimodal adapter in the visual branch becomes higher, which demonstrates the ability of our *AVMoE* to dynamically combine adapters according to different scenarios.

In addition, we observe that when the model is processing visual-only data, the activation probability of the unimodal adapter in the visual branch becomes higher, which proves the ability of our *AVMoE* to dynamically activate adapters according to different scenarios.

# F    Feature Visualization

Figure 8 visualizes the learned audio and visual features by employing t-SNE [55]. Each point represents the feature of an individual audio or visual event, with each color indicating a specific category. The visualizations reveal that the features extracted by the proposed *AVMoE* model are more compact within classes and more distinct between classes, which demonstrates that our *AVMoE* can learn discriminative features for each modality across various audio-visual downstream tasks.

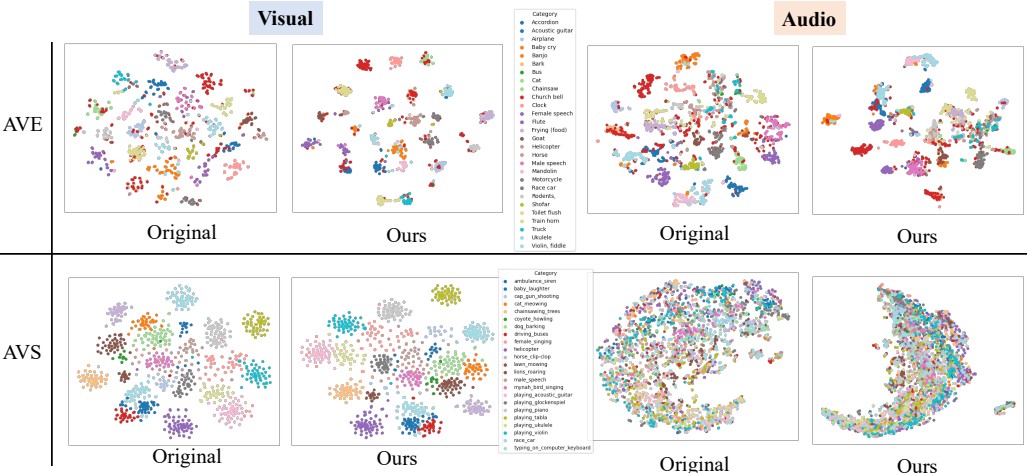

Figure 8: Qualitative visualizations of visual and audio features of original and Ours on AVE and AVS tasks.

