# OpenReview forum: "Mixtures of Experts for Audio-Visual Learning"
_NeurIPS.cc/2024/Conference — NeurIPS 2024 poster_

### Official Review · Reviewer_usTX · 2024-06-27

**Soundness:** 3
**Presentation:** 3
**Contribution:** 3
**Rating:** 7
**Confidence:** 4

**Summary:**

They propose a novel and simple method based on Mixture of Experts (MoE) for audio-visual learning.

**Strengths:**

- Simple yet effective method.
- State-of-the-art results.
- Overall, is well written.

**Weaknesses:**

- Does not include demos with audio to listen in a website.
- Does not include code.

Minor notes on the writing:
- Mention AVE, AVVP, AVS, AVQA in the abstract without definition.
- Section 2.1 first paragraph references to "early researchers" without citations.
- line 82: typo missing closing ).
- Section 3 there is no space before parenthesis.
- line 232: space missing after the parenthesis.
- I did not understand how the unimodal adapter works. I have the feeling that cross-modal adapter is well explained, but the unimodal adapter it is not. Maybe figures for how they work would be useful.

**Questions:**

- Can you include demos and code?

**Limitations:**

The evaluation framework is limited to the evaluation tasks the authors have chosen.

---

> ### Author Rebuttal · Authors · 2024-08-07
>
> Dear reviewer usTX,
>
> Thank you for the positive feedback and valuable suggestions! Please see the following for our point-by-point reply.
>
> ---
> **Weakness - Demos**
>
> Thanks for your advice! We totally agree with the importance of providing audio-visual demos for a more comprehensive understanding of our work, and we will build a dedicated project page to provide source codes and demos.
>
> ---
> **Weakness - Code**
>
> Thanks for your suggestion! We have provided our code in the supplementary materials. We will release our source codes and provide more implementation details on the project website.
>
> ---
> **Weakness - Minor Typos**
>
> Thanks for your careful review! We have corrected the typos and revised our manuscript. Besides, the architectures of cross-modal and unimodal adapters are shown in Appendix E, and we will consider moving this section to the main text for better clarity.

---

### Official Review · Reviewer_k9Vn · 2024-07-22

**Soundness:** 2
**Presentation:** 3
**Contribution:** 1
**Rating:** 5
**Confidence:** 5

**Summary:**

This paper introduces an interesting observation i.e. for the off-screen sound cases or some mixed audio cases, ``injecting cross-modal adapters only brings disturbing information``.  To solve the problem. this paper proposes a Mixture of expert strategy that can let the model focus on different aspects of the input data, ignoring the interference. However, I found that the technique's novelty is limited, it seems just have little modification based on [1], just the input audio-visual tokens are first compressed via a cross-attention operation. Moreover, The improvements this approach brings to downstream tasks are also limited, for the AVEL task, compared to DG-SCT, there is just a 0.4% improvement on ACC. For AVS, there is just a 0.2% improvement on $M_j$ for the S4 setting, and so on.
[1] Vision Transformers are Parameter-Efficient Audio-Visual Learners

**Strengths:**

This paper presents an interesting problem that the off-screen sound or the mixture audio signals may hinder the effectiveness of the audio-visual interactions based on cross-attention.

**Weaknesses:**

1. The technique novelty is limited, there is no obvious difference between this method design and [1]. Moreover, the architecture are same as [2].  Figure 7 is totally the same as Figure 2 in [1].
[1] Vision Transformers are Parameter-Efficient Audio-Visual Learners
[2] Cross-modal Prompts: Adapting Large Pre-trained Models for Audio-Visual Downstream Tasks
2. This paper gives the off-screen and mixed sound cases to illustrate their motivation, but in the experiment part, they did not present the final results of these cases.

**Questions:**

Please see the weakness.

**Limitations:**

1. Limited technique innovations.
2. Lacking experiments to demonstrate the effectiveness of the proposed method.

---

> ### Author Rebuttal · Authors · 2024-08-07
>
> Dear reviewer k9Vn,
>
> Thank you for taking the time to read our paper. We hope our responses adequately addresses your concerns.
>
> ---
> **Summary - The improvements this approach brings to downstream tasks are limited.**
>
> Thanks for your comments! However, we argue that focusing only on some of the experimental results might be partial.
> Firstly, it is important to note that our method achieves better results with fewer parameters compared to DG-SCT. While DG-SCT incorporates a more complex design for the adapter structure, we adopt a standard bottleneck structure of adapter, similar to that in LAVisH.
>
> Secondly, regarding more challenging scenarios like the MS3 setting of AVS task (multi-sound sources), our AVMoE using fewer parameters improves by 1.0% on $\mathcal{M_J}$ and 4.5% on $\mathcal{M_F}$ compared with DG-SCT. Besides, for more complex tasks like AVQA and AVVP, our AVMoE improves by 0.9% ~ 2.0% compared with DG-SCT. It can be viewed that AVMoE achieves significant performance across multiple audio-visual tasks, which has also been acknowledged by other reviewers.
>
> ---
> **Weakness - Limited technique Novelty**
>
> Firstly, LAVisH and DG-SCT only inject cross-modal adapters into frozen audio-visual pre-trained models. LAVisH and DG-SCT only focus on cross-modal adapters and ignore the intra-modal information, and only introducing cross-modal adapters may bring irrelevant information in some challenging cases. To dynamically introduce inter-modal and intra-modal information, we propose an approach of Mixture of Experts (MoE) that combines cross-modal and unimodal adapters according to different scenarios. **The MoE strategy for audio-visual learning is our main contribution, rather than input audio-visual tokens of adapters.** Regarding your concerns about the architecture of the entire model, the overall framework of multimodal parameter-efficient transfer learning methods all consists of two frozen backbones and trainable modules, and our proposed AVMoE module is illustrated on the right side of Figure 2.
>
> Secondly, as for the adapter architecture in Figure 7, this is the typical bottleneck architecture of adapters, which is always used in adapter-related works (like the transformer architecture in many works). This is not our main contribution, but in order to facilitate readers who are not in this field to have a clearer understanding of adapters, we present the details in Appendix E. Besides, we also mentioned in the main text (line 143) that the adapter structure is based on LAVisH.
>
> ---
> **Weakness - Effectiveness of the Proposed Method.**
>
> Thanks for your suggestion! Yes, Figure 1 illustrates our motivation. However, this is the example in the AVVP task, which has more labels of audio and visual events. Due to space limitations, we present the qualitative and quantitative experiments of the AVVP task in Appendix D.
> Given the limited time available for finalizing the appendix, Figure 6 may not fully illustrate the complexity of audio-visual scenarios. We apologize for this and show a more challenging example in Figure 2 of the response PDF.
>
> In this example, the video label only includes *car*, and the audio label includes *car* and *speech*, which are not exactly the same. We observe that the visual predictions of DG-SCT are affected by the audio features, resulting in a visual prediction of *speech*, which does not exist in the video frame. On the other hand, our AVMoE retains unimodal information and has MoE modules to adjust the weight distribution for audio-visual features. This avoids the interference of inconsistent information and obtains the correct result. As for the audio predictions, our model can better localize the temporal boundaries of events compared to DG-SCT, which reveals that our model is capable of dealing with complex scenarios.

---

### Official Review · Reviewer_YVYv · 2024-07-23

**Soundness:** 3
**Presentation:** 2
**Contribution:** 3
**Rating:** 6
**Confidence:** 5

**Summary:**

The manuscript proposes the Audio-Visual Mixture of Experts (AVMoE) approach for multiple audio-visual tasks, aiming to explore parameter-efficient transfer learning for audio-visual learning. Specifically, AVMoE introduces both unimodal and cross-modal adapters as multiple experts, specializing in intra-modal and inter-modal information respectively. A lightweight router dynamically assigns the weights of each expert based on the specific requirements of each task. The effectiveness of the proposed method is demonstrated across multiple task datasets.

**Strengths:**

1. The writing is easy to read.
2. The proposed AVMoE shows performance improvements across multiple audio-visual tasks.

**Weaknesses:**

1. The AVMoE is somewhat similar to LAVISH, and it would be beneficial to discuss the similarities and differences between the two works.
2. Using all audio and visual tokens as latent tokens can be very computationally expensive.
3. In Table 3, note that the Visual Encoder for both AVSD and Pano-AVQA methods uses the ImageNet pre-trained ResNet-18 model, not VGG-19 and Faster RCNN.
4. Although the experimental results demonstrate the method's effectiveness, does this effectiveness truly capture audio-visual scenarios well? The submitted manuscript should also include temporal and spatial visualizations for the AVQA task.
5. It is recommended to include a more comprehensive comparison of methods in the experiments, especially for the AVQA and AVS tasks, as many recent works have focused on them.
6. The effectiveness of the Router in the method needs to be validated.
7. The rationale for setting the batch-size to 1 requires further discussion.
8. Some writing conventions need attention, eg. the LAVish reference on line 250.
9. Providing the code would be beneficial.

**Questions:**

If the author can clarify the above questions or suggestions, I will consider raising the score accordingly.

**Limitations:**

The authors adequately addressed the limitations and potential negative societal impact of their work.

---

> ### Author Rebuttal · Authors · 2024-08-07
>
> Dear reviewer YVYv,
>
> Thank you so much for these very valuable and constructive suggestions! Please kindly find the point-to-point responses below.
>
> ---
> **Differences between AVMoE and LAVisH**
>
> Thanks for your feedback! The similarity between AVMoE and LAVisH is the bottleneck architecture of adapters. However, LAVisH only introduces cross-modal adapters, whereas we aim to design a more flexible approach to combine the capabilities of different adapters. Therefore, we introduce cross-modal and unimodal adapters synergistically and design a router to dynamically activate adapters according to different scenarios, thereby alleviating potential conflicting information in multimodal data. We will include these analyses in the main text to better clarify the differences between AVMoE and LAVisH.
>
>
> ---
>
> **Latent Tokens**
>
> There might be some misunderstanding. As we mentioned in Line 146, latent tokens are small, randomly initialized vectors for compressing audio and visual tokens to reduce the computational complexity. Audio and visual tokens are not latent tokens. We will refine this part to make it clearer.
>
>
>
>
> ---
> **Visual Encoder of AVQA Methods**
>
> Thanks for your comment! We have double-checked the original paper of AVSD [A] and Pano-AVQA [B]. AVSD utilizes VGG-19 as visual encoder and compares it with I3D visual features. Pano-AVQA indeed uses Faster R-CNN as visual encoder.
>
> AVSD: In the Sec. 2 of [A], "we assess a naive feature extractor based on **VGG**, and demonstrate that for video- reasoning, careful reduction of the spatial dimension is more crucial than the type of extracted features used to embed the video frames."
>
> Pano-AVQA: In the Sec. 4.1 of [B], "we use **faster R-CNN** trained with ImageNet Detection to extract and represent region proposals".
>
> *[A] Schwartz I, Schwing A G, Hazan T. A simple baseline for audio-visual scene-aware dialog, CVPR 2019.*
>
> *[B] Yun H, Yu Y, Yang W, et al. Pano-avqa: Grounded audio-visual question answering on 360deg videos, ICCV 2021.*
>
> ---
>
> **Temporal-Spatial Visualizations for AVQA**
>
> Thanks for your suggestion! We have already conducted experiments and provided visualizations for the AVS task, which demonstrate that our method effectively captures the correlation between audio and visual elements. Therefore, we did not include temporal-spatial visualizations for the AVQA task in the submitted manuscript. However, we appreciate your suggestion and present the visualizations of the AVQA task in Figure 1 of the response PDF.
> It can be seen that our AVMoE localizes the instruments more concentratedly and more precisely. In the first case that contains the question of left and right instruments, our AVMoE assigns greater weight to the instruments on the left and right accordingly, whereas DG-SCT appears to distribute attention across all three instruments. In the second case, our AVMoE can even completely localize the entire cello and accordion, whereas DG-SCT mistakenly localizes to the stand rather than the instruments in the last frame.
>
>
> ---
> **More Comparisons for the AVQA and AVS tasks**
>
> Due to space limitations, our work focuses on the methods of audio-visual parameter-efficient transfer learning. Following LAVisH and DG-SCT, we choose the same methods of AVS and AVQA for comparison. However, we also agree that adding more comparisons would help understand the progress of research tasks, and we present the comparison results in the Table 2-4 of the response PDF. We observe that there still exists the gaps compared to task-specific methods, but our AVMoE can reduce these gaps.
>
> ---
> **The Effectiveness of the Router**
>
> Thanks for your advice! The activation probability of experts in Figure 4 of the original manuscript demonstrates that the router can dynamically activate adapters according to different scenarios. Besides, we also conduct additional experiments to illustrate the effectiveness of the router quantitatively, and the experimental results are shown in Table 1 of the response PDF. The comparison between Setting #1 and #2 demonstrates the importance of routers in our model.
>
> ---
> **Batchsize**
>
> Due to the limitations of computational resources, the batch size on a single GPU is set to 1 for AVE and AVVP tasks. However, to mitigate the limitations of using a small batch size, we employed a gradient accumulation strategy, which is a common technique for training large models with limited GPU memory. This approach involves accumulating gradients from multiple mini-batches before performing a weight update. By doing so, we can effectively simulate a larger batch size (the same as other models).
>
>
> ---
> **Writing**
>
> Thanks for pointing this out! We have carefully checked the manuscript and corrected the typos.
>
>
> ---
> **Code**
>
> Thanks for your advice! Actually, we have provided the code in supplementary materials. We will release our source codes and provide more implementation details on GitHub.

---

> > ### Author Response · Authors · 2024-08-14
> >
> > Dear reviewer YVYv,
> >
> > Thank you for your valuable reviews! We are not sure whether our previous response has adequately addressed your concerns. If you have any further questions or comments, please kindly let us know, and we will be glad to respond.

---

### Official Review · Reviewer_jXV1 · 2024-07-27

**Soundness:** 3
**Presentation:** 3
**Contribution:** 3
**Rating:** 7
**Confidence:** 4

**Summary:**

This paper proposed an integrated adaptor architecture that makes use of cross-modal attention mechanism and Mixture-of-Experts. The architecture connects (separately) trained audio and visual encoder to do audio-visual tasks, including audio-visual event localization, audio-visual segmentation, audio-visual question answering, and audio-visual video parsing. Experiments show that the proposed architecture outperforms other adaptors while having less tunable parameters.

**Strengths:**

The method is straight forward and the paper is well-written. The introduced architecture is novel and intuitive. The improvements over other methods in terms of parameter count and efficiency is clear.

**Weaknesses:**

The comparison with other approaches stays at parameter count and task performance, but inference cost, training time, and flops are not reported.

How does the approach compare with SotA on these datasets (rather than just adaptor models)?

It would be helpful if the author could provide some explanation on why the proposed approach is better than DG-SCT and LAVisH.

The main text and appendix are disconnected, it would be helpful to put some pointers in the main text which refers to appendix.

**Questions:**

I wonder why is audio visual event classification not compared, it's also not compared in LAVisH and DG-SCT, so I wouldn't blame the author for that, but I'm curious why such an common an popular audio visual task is not considered in audio visual adaptor papers.

**Limitations:**

The approach is only for audio visual tasks (but not other joint modal modeling scenarios) and the amount of tasks considered is limited.

---

> ### Author Rebuttal · Authors · 2024-08-07
>
> Dear Reviewer jXV1,
>
> Thanks for your careful review and kindly comments on this paper. Please see our point-by-point responses below.
>
>
> ---
> **Weakness  - The inference cost, training time, and flops are not reported.**
>
> Thanks for your suggestion! Following LAVisH and DG-SCT, we only report the trainable and total parameters, which can also demonstrate the efficiency of our approach. In order to provide a more comprehensive comparison of the efficiency, we will add these metrics in the Appendix.
>
> | **Method** | **Inference cost** | **Train Time** | **GFLOPS** | **Total Params↓** |  **Acc** |
> |:----------:|:------------------:|:--------------:|:----------:|:-----------------:|:--------:|
> |LAVisH|0.13s|7h|406.7|238.8M|81.1|
> |DG-SCT|0.13s|15h|460.8|460.8M|82.2|
> |**AVMoE**|0.13s|12h|422.6|404.0M|**82.6**|
>
> **Inference Cost: The average time it takes for each batch to be processed by the model on a NVIDIA GeForce RTX 3090.*
>
> **Since these two methods have not released this information, their inference costs and training times are calculated by our runnings of their open source codes.*
>
> ---
> **Weakness - Comparison with SOTA on these datasets (rather than just adaptor models).**
>
> Thanks for your question. We have reviewed recent related works and reported their results in Table 2-4 of the response pdf. It should be noted that these excellent related works [A, B, E, F, G, H, I, J] are specially designed according to the characteristics of these tasks and datasets. When comparing audio-visual parameter-efficient transfer learning methods ("General Methods" in the tables) with them, we observe that there still exist gaps, but our AVMoE can help to reduce the remaining gaps.
>
> *[A] Learning Event-Specific Localization Preferences for Audio-Visual Event Localization, ACM MM 2023.*
>
> *[B] CACE-Net: Co-guidance Attention and Contrastive Enhancement for Effective Audio-Visual Event Localization, ACM MM 2024.*
>
> *[C] Vision Transformers are Parameter-Efficient Audio-Visual Learners, CVPR 2023.*
>
> *[D] Cross-modal prompts: Adapting large pre-trained models for audio-visual downstream tasks, NeurIPS 2024.*
>
> *[E] Stepping stones: A progressive training strategy for audio-visual semantic segmentation, ECCV 2024.*
>
> *[F] Can Textual Semantics Mitigate Sounding Object Segmentation Preference?, ECCV 2024.*
>
> *[G] Unveiling and Mitigating Bias in Audio Visual Segmentation, ACM MM 2024.*
>
> *[H] CAD-contextual multi-modal alignment for dynamic AVQA, WACV 2024.*
>
> *[I] Semantic Enrichment for Video Question Answering with Gated Graph Neural Networks, ICASSP 2024.*
>
> *[J] Parameter-Efficient Transfer Learning for Audio-Visual-Language Tasks, ACM MM 2023.*
>
> ---
> **Weakness - More explanation of why AVMoE is better than DG-SCT and LAVisH.**
>
> Thanks for your advice! We will analyze more about the advantages of our method compared with LAVisH and DG-SCT in the manuscript. LAVisH and DG-SCT only inject cross-modal adapters into frozen audio-visual pre-trained models. LAVisH adopts the typical bottleneck architecture of adapters, whereas DG-SCT designs a more complex architecture of dual-guided spatial-channel-temporal attention mechanism in adapters. However, they only focus on cross-modal adapters and ingore the intra-modal information, and only introducing cross-modal adapters may bring irrelevant information in some challenging cases. Hence, to dynamically introduce inter-modal and intra-modal information, we propose an approach of Mixture of Experts (MoE) that combines adapters according to different scenarios.
>
>
> ---
> **Weakness - The main text and appendix are disconnected.**
>
> Thanks for your valuable comments! We will include more pointers in the main text.
>
>
> ---
> **Questions - No mention of the audio-visual event classification task.**
>
> Thanks for your question! We are not sure if you are asking about the audio-visual classification task on datasets such as VGGsound [K]. Although the VGGsound dataset contains a large number of videos and rich types of annotations, it only has coarse video-level annotations. As far as we know, the VGGsound dataset is more commonly used for pre-training. The AVE and AVVP tasks, which involve classifying audio-visual events second-by-second and localizing their boundaries, are more challenging and provide a more rigorous evaluation of the model's performance.
>
> *[K] Vggsound: A large-scale audio-visual dataset, ICASSP 2020.*

---

> > ### Comment · Reviewer_jXV1 · 2024-08-12
> > **Thanks for the reply**
> >
> > Thanks for taking the time to reply to my review! I'll keep my rating.

---

> > > ### Author Response · Authors · 2024-08-12
> > >
> > > Thanks for your reply! We are glad that you keep the original positive rating.

---

### Official Review · Reviewer_pdXh · 2024-07-29

**Soundness:** 2
**Presentation:** 2
**Contribution:** 3
**Rating:** 7
**Confidence:** 4

**Summary:**

This paper proposes AVMoE, a new way of adding adapters to pretrained audio-visual models.

AVMoE extends the LAVisH approach in two ways:
(1) Using unimodal adapters in addition to cross-modal LAVisH adapters.
(2) Adding a router network to dynamically combine the output of multiple unimodal/cross-modal adapters.

AV-MoE outperforms two existing methods (LAVisH and DG-SCT) on three audio-visual tasks: event localization, segmentation, and question answering.

**Strengths:**

Results show solid improvements:
- Extensive experiments demonstrate that AVMoE outperforms the existing DG-SCT approach on all three tasks, while using less trainable parameters.
- Modality ablation studies clearly show that AVMoE also handles missing audio modality much better than DG-SCT, which is a much desired property for audio-visual models.

**Weaknesses:**

### Experimental Design Ablation

- It is less clear how much each component of the proposed method contributes to the improvement over LAVisH.

  What if only unimodal adapters were added, without the router? What if we only add the MoE router to 2 or more cross-modal LAVisH adapters?
  Additionally, experiments with adjusted LAVisH hyperparameters to increase trainable parameters to better compare with AVMoE could also be insightful.

### Unclear how hyperparameters are chosen

- The appendix uses completely different num. of latent tokens and learning rate compared to LAVisH, without justifying (1) how they were chosen and (2) how much they affect the results.


### Presentation
- Some important details of the proposed method are either left in the appendix (adapter architecture) or omitted entirely (what is the variance of Gaussian noise used? How much does it affect the results?).

  I would suggest moving non-essential results (Results of all systems using incomparable encoders, Qualitative examples of Figure 3, etc.) to the appendix and moving the description of important details such as adapter architecture to the main text.
- Minor comments:
  1. **The references section contains multiple mistakes.** Some papers are cited more than once, some lack venue titles, and some NeurIPS 2023 papers are cited as NeurIPS 2024.
  2. I would consider prior works in multimodal parameter-efficient transfer learning to be much more relevant than MoE. It would be nice to see some discussion of parameter-efficient methods in vision-and-language (A), audio-visual (B, or even concurrent works such as C) beyond what was used in the paper.
  3. Regarding MoE related work, Akbari et al. also work on audio-visual inputs.
  4.  Qualitative examples of the M3 setting in Figure 3 are a bit hard to see with tiny white masks.

[A] Yi-Lin Sung, Jaemin Cho, and Mohit Bansal. Vl-adapter: Parameter-efficient transfer learning for vision-and-language tasks. In CVPR, 2022

[B] Liu, Hongye, et al. "Parameter-Efficient Transfer Learning for Audio-Visual-Language Tasks." Proceedings of the 31st ACM International Conference on Multimedia. 2023.

[C] Kai Wang, Yapeng Tian, Dimitrios Hatzinakos. Towards Efficient Audio-Visual Learners via Empowering Pre-trained Vision Transformers with Cross-Modal Adaptation. In CVPR, 2024

**Questions:**

- When comparing results with shared Swin-V2-L encoder versus using seperate Swin-V2-L & HTS-AT encoders, why does LAVisH performance degrade whereas AVMoE improves?
- What is the meaning behind the asterisk in LAVisH row of Table 3: Audio-Visual Question Answering?

**Limitations:**

As mentioned in the Weaknesses section:
1. Importance of each component of proposed method is not well studied,
2. The choice/effect of different hyperparameters compared to LAVisH is not justified,
3. Method details such as model architecture are left in the appendix or omitted entirely.

However, AVMoE seems to shows strong improvements over existing approaches. If the proposed method consistently excels across hyperparameters, and the method section is fleshed out, I would be inclined to accept the paper.

---

> ### Author Rebuttal · Authors · 2024-08-07
>
> Dear reviewer pdXh,
>
> Thank you so much for these very insightful and constructive comments, please see the following for our point-by-point responses.
>
> ---
> **Weakness - Experimental Design Ablation**
>
> Following your great advice, we conducted more experiments to analyze the contribution of each component, and the results are shown in Table 1 of the response PDF.
>
> - **Effect of Routers**: The comparison between Setting #1 and #2 reveals a consistent performance drop across all tasks when routers are not included in the model. This result highlights the importance of routers in our approach.
>
> - **Effect of only adding more Cross-modal Adapters without Unimodal Adapters**: The results between Setting #3 and #4 illustrate that adding more CMAs without UAs does not significantly improve the model's performance. However, when we compare Setting #4 to Setting #1, there is a noticeable decrease in performance. This result highlights the importance of incorporating UAs into the model, which demonstrates that a balance between unimodal and cross-modal adapters is crucial in our model to effectively capture both intra-modal and inter-modal information.
>
> - **Adjustment of LAVisH Hyperparameters**: We have attempted to adjust LAVisH hyperparameters, such as downsample size and latent token number, but could not match AVMoE's parameter size. Moreover, the comparison between DG-SCT and AVMoE in the manuscript shows that our AVMoE can achieve significant performance with fewer parameters. These comparison results illustrate our approach's superior parameter efficiency and effectiveness.
>
>
> **Weakness- Hyperparameters**
> - The number of latent tokens: Our hyperparameters of latent tokens and learning rates in these audio-visual downstream tasks are based on DG-SCT. From the experimental results in the table below, it can be viewed that the number of latent tokens has little effect on the total parameters and performance of models. Hence, for comparison with DG-SCT, we follow their hyperparameters and set the number of latent tokens for AVE and AVVP tasks to 32. Moreover, even if we use the same number of latent tokens as LAVisH, the performance of our model is still better than LAVisH.
>
>     **AVE:**
>     | **Laten Tokens** | **Total Params** | **Acc** | **LAVisH Acc** |
>     |---|---|---|---|
>     | 2 | 403.7M | 82.2% | 81.1% |
>     | 8 | 403.8M | 82.3% | - |
>     | 32 | 404.0M | **82.6%** | - |
>
>     **AVS:**
>     | **Laten Tokens** | **Total Params** | **S4** |  | **MS3** |  |
>     |---|---|---|---|---|---|
>     |  |  | $\mathcal{M_J}$ | $\mathcal{M_F}$ | $\mathcal{M_J}$ | $\mathcal{M_F}$ |
>     | 2 | 501.2M | **81.1** | 89.7 | **54.5** | **68.7** |
>     | 8 | 501.6M | 80.8 | 89.7 | 54.3 | 68.5 |
>     | 32 | 503.0M | 80.9 | **89.9** | 54.2 | **68.7** |
>
>     **AVQA:**
>     | **Laten Tokens** | **Total Params** | **Avg Acc** |
>     |---|---|---|
>     | 2 | 456.6M | **75.7** |
>     | 8 | 456.7M | 75.6 |
>     | 32 | 456.9M | **75.7** |
>
>
>
> **Weakness - Presentation**
>
> * Some details of architecture: Thanks for your constructive comments! We will adjust the manuscript structure based on your suggestion.
> * The detail and effect of Gaussian Noise: Adding noise (e.g., Gaussian noise) to MoE in the training stage is very important, which can prevent the model from over-relying on certain experts. The Gaussian noise is implemented with a standard normal distribution (mean 0 and variance 1). We conducted an additional ablation study to evaluate the effect of Gaussian noise. The results indicate that incorporating Gaussian noise not only improved the model's performance on various datasets but also contributed to the training stability of the MoE model with less standard deviations.
>
>     | Num. | Settings | AVE (Acc. ± Std Dev) | AVS-S4 ($\mathcal{M_J}$ ± Std Dev) | AVQA (Acc. ± Std Dev) |
>     |---|---|---|---|---|
>     | #1 | AVMoE (full model) | 82.6 ± 0.2 | 89.7 ± 0.3 | 75.7 ± 0.2 |
>     | #2 | AVMoE w/o Gaussian Noise | 81.5 ± 1.3 | 88.4 ± 1.1 | 74.4 ± 0.9 |
>
> * References: We copied the bibtex directly from Google Scholar without making any changes, maybe there was something wrong. We have carefully checked these mistakes and corrected them.
> * Related work: We totally agree that parameter-efficient transfer learning is more relevant to our work. However, Mixture of Experts (MoE) is our main contribution, we present a separate subsection (Sec. 2.2) for MoE and discuss audio-visual parameter-efficient methods in the subsection of audio-visual learning (Sec. 2.1). But thanks for your suggestion, we will add a subsection for discussing multimodal parameter-efficient methods.
> * Clarity of Qualitative examples: Thanks for pointing this out! Actually, we have tried two methods to display the qualitative results of AVS: adding white masks to original images and directly showing the binary masks.
> We thought the former one could better demonstrate the masks of semantic objects, but the white masks are indeed not clear enough. We will consider the second plan and add bounding boxes on important areas for better comparison.
>
> ---
> **Question 1 - Why does LAVisH performance degrade whereas AVMoE improves when using separate Swin-V2-L and HTS-AT encoders?**
>
> The AVE performance of LAVisH with separate Swin-V2-L and HTS-AT encoders in our Table 1, is referenced from Table 1 of the DG-SCT paper. The authors of DG-SCT mentioned that the reason why the LAVisH dropped maybe it only utilizes latent tokens for cross-attention. The coarse extracted cross-modal information may not adequately counteract the negative effects of domain gaps introduced by the use of different encoders. We also argue that audio representations may bring disturbing information sometimes, and our AVMoE is designed to alleviate this problem.
>
> **Question 2 - What is the meaning behind the asterisk in LAVisH row of Table 3?**
>
> LAVisH* denotes our implementation version of LAVisH. We will add this annotation to the caption of Table 3.

---

> ### Comment · Reviewer_pdXh · 2024-08-12
>
> Thank you for adding comprehensive ablations studies.
>
> ### **Experimental Design Ablation**
> Extensive results show that each component of the proposed method is beneficial. I encourage the authors to add these results to the Table 4 in the main paper.
>
> ### **Hyperparameter selection**
> The ablation experiment on num. of latent tokens is greatly appreciated. Results show that AVMoE is robust to the num. of latent tokens.
> However, the learning rate and batch size seems to be different from both LAVisH and DG-SCT? To quote each paper on hyperparams for the AVE task:
> > LAVisH: We set the learning rate of LAVISH adapter to 5e−6, and 4e−6 for the final prediction layer for audio-visual event localization, ...
> | Task | Batch Size |
> --- | ---
> | AVE | 2 |
>
> > DG-SCT: For the AVE ... We train the model with a batch size of 8 and a learning rate of $5 × 10^{−4}$ ...
>
> > AVMoE: ... weight updates after every 16 batch of training ... We set the learning rate of the AVMoE adapter to 5e-4, while the learning rate of the final prediction layer is determined by the task, 5e-6 for AVE ...
>
> ### **Presentation**
> Thank you for showing the importance of Gaussian noise for the router. Regarding the references, [2], [10], [24] & [25], [38], [42] and [43], [60] and [61] are the errors that stand out the most.
>
> ### **Questions**
> If Swin-V2-L + HTS-AT is reimplemented, the relevant rows in Table 1 and Table 2 probably should be marked as well.
>
>
> Overall, while hyperparameter selection details are a bit vague, the extensive experiments in the rebuttal has addressed most of my major concerns. I would like to change my rating from 4 to 7.

---

> > ### Author Response · Authors · 2024-08-12
> >
> > Dear reviewer pdXh,
> >
> > We sincerely thank you for your thorough reviews and valuable suggestions!
> >
> > ---
> > **Hyperparameter Selection**
> >
> > We have double-checked the source code, and confirmed that our learning rates are the same as those used in DG-SCT.
> > * For the AVE task, the learning rate of DG-SCT adapter is also 5e−4, and 5e−6 for the final prediction layer (not mentioned in DG-SCT paper).
> > * For the AVS task, the learning rate is 3e−4 for the S4 setting and 1.5e−4 for the MS3 setting, consistent with DG-SCT. Sorry for missing the learning rate of S4 setting in the original manuscript.
> > * For the AVVP task, the learning rate is 3e−4, we apologize for this typo and will revise the learning rate of AVVP task and AVS S4 setting in the next version.
> >
> > Regarding the batch size, we can only use a smaller batch size due to computational resource limitations. However, we employ a gradient accumulation strategy to mitigate the limitations, which is a common technique for training large models with limited GPU memory.
> >
> > **Presentation**
> >
> > We would like to extend our sincere appreciation for your effort in carefully reviewing our work! We have carefully checked these references & tables and will update them in the next reversion.

---

### Author Rebuttal · Authors · 2024-08-07

We sincerely thank all reviewers for your valuable and constructive comments. Our motivation is to propose a general, efficient, and flexible method for audio-visual learning, and your suggestions will definitely help us strengthen our work. Please kindly find the point-to-point responses below.

---

> ### Author Response · Authors · 2024-08-12
>
> Dear reviewers,
>
> Thank you so much for reviewing our paper and for your valuable suggestions. We hope our responses have addressed your concerns. If you have any further questions or comments, please kindly let us know, and we will be glad to respond.

---

### Decision · Program_Chairs · 2024-09-25

**Decision:**

Accept (poster)

**Comment:**

This paper proposes AVMoE (Audio-Visual Mixture of Experts), a new way of adding adapters to pretrained audio-visual models.
AVMoE extends the LAVisH approach in two ways: (1) Using unimodal adapters in addition to cross-modal LAVisH adapters.   (2) Adding a router network to dynamically combine the output of multiple unimodal/cross-modal adapters. AV-MoE outperforms two existing methods (LAVisH and DG-SCT) on audio-visual tasks including audio-visual event localization, audio-visual segmentation, audio-visual question answering, and audio-visual video parsing.

It is novel and intuitive to leverage MoE in this problem space, to learn both inter- and intra- modality information. Experiment results also support the efficacy of this approach; experiment design is comprehensive and sound. Thus I recommend to accept this paper.

Strength
- Paper is well written.
- Proposed architecture is novel and intuitive.
- Solid improvements: extensive experiments demonstrate that AVMoE outperforms the existing DG-SCT approach on all three tasks, while using less trainable parameters. Modality ablation studies clearly show that AVMoE also handles missing audio modality much better than DG-SCT, which is a much desired property for audio-visual models.


Weakness
- Some gap compared with SOTA (not just adaptor models) on evaluated audio-visual tasks. This is understandable as most SOTA models are designed specifically for each task. We would love to see authors to explore the combination of proposed method to existing approaches for any further improvement.
- Some reviewers pointed out that AVMoE is similar to LAVisH and thus marginally novelty. This is a reasonable argument. I think every work is built upon previous literatures and it is inevitable to have overlapping. Seems to me MoE is the core contribution of this work and using MoE is well motivated in this problem space (leverage MoE to learn both inter- and intra- modality information), and I can see the potential for proposed approach to be generalizable to other similar problems. Thus I think this paper is novel and important enough for NeurIPS community.